



# Tomographic Imaging of a Large Scale TID during the Halloween Storm of 2003

Karl Bolmgren[1], Cathryn Mitchell[1], Talini Pinto Jayawardena[1], Gary Bust[2], and Jon Bruno[1]

[1]Department of Electronic and Electrical Engineering, University of Bath, UK
[2]Johns Hopkins University Applied Physics Laboratory, MD, USA

**Correspondence:** Karl Bolmgren (k.h.a.bolmgren@bath.ac.uk)

**Abstract.** The most intense ionospheric storm observed in recent times occurred between 29-31 October 2003. The disturbances to the high-latitude regions set off several Large-Scale Travelling Ionospheric Disturbances (LSTIDs), wavelike perturbations in the ionospheric electron density. This paper investigates one particular Travelling Ionospheric Disturbance (TID) on 31 October 2003 using North American Global Positioning System (GPS) receiver network data and a tomographic imaging

technique. The TID has an estimated period of 30 min, an estimated horizontal wavelength of 700 km and propagates South-Westward over North America. The tomographic reconstruction of the wave is validated using a simulation of the observations and with independent observations from ionosondes and the CHAMP Planar Langmuir Probe. The results are discussed in the context of the magnetic and ionospheric conditions that may have contributed to the launch of the wave. Large-scale TIDs are challenging to study over large regions of the Earth, and the GPS network here is shown to offer a unique perspective on the

spatial and temporal variation of the TID. The experimental results are backed up by simulations that show a denser network of receivers, as is available in more recent years, would produce improved accuracy in the TID imaging.

## 1 Introduction

Travelling Ionospheric Disturbances (TIDs) are ionospheric manifestations of Atmospheric Gravity Waves (AGWs) occurring in the neutral atmosphere (Hines, 1960). AGWs are buoyancy waves in the atmosphere, and can be observed as TIDs when

they transfer momentum to ions in the ionosphere by collision. Large Scale TIDs (LSTIDs) are a common occurrence during geomagnetic storms. LSTIDs are wavelike perturbations in the ionospheric electron density with typical wavelengths over 1000 km, periods between 0.5-3 h (Hocke and Schlegel, 1996) and typically travelling equatorwards from the auroral regions (Davis and da Rosa, 1969). LSTIDs perturb the electron density and hence the Total Electron Content (TEC), the number of free electrons along a path through the ionosphere, on scales up to several TEC units (1 TECu = $10^{16}$ free electrons per m$^2$).

TEC is proportional to the first order ionospheric delay of transionospheric radio waves propagating in the ionosphere, and is therefore a crucial parameter for Global Navigation Satellite Systems (GNSS).

Between 29-31 October 2003 a series of large Coronal Mass Ejections (CMEs) – expulsions of plasma from the solar corona – travelling along the interplanetary magnetic field reached the magnetosphere of the Earth, causing strong geomagnetic storms. These are often referred to as the *Halloween Storm(s) of 2003*. The CMEs caused two sudden storm onsets on 29 October 2003





and 30 October 2003 (e.g. Mannucci et al., 2005; Horvath and Lovell, 2010). The planetary K-index (Kp) peaked at 9 on 29 and 30 October 2003, and 8 on 31 October. Kp remained above 4 throughout 31 October, which, although still disturbed, constituted the recovery phase corresponding to the second sudden onset. The Auroral Electrojet index (AE) reached a maximum of 1827 nT at 06:31 UTC on 31 October, which is plotted in Figure 1. Change in AE is related to auroral ionospheric current activity, which has been correlated with the appearance of TIDs at mid-latitudes (Hajkowicz, 1991; Hunsucker, 1982; Hocke and

Schlegel, 1996; Lewis et al., 1996). These TIDs are thought to be launched by Joule heating of the atmosphere caused by increased ionospheric currents. High variability in AE occurred several times throughout 31 October, as seen in Figure 1. This variability in AE provides evidence for a potential TID generation mechanism being present.

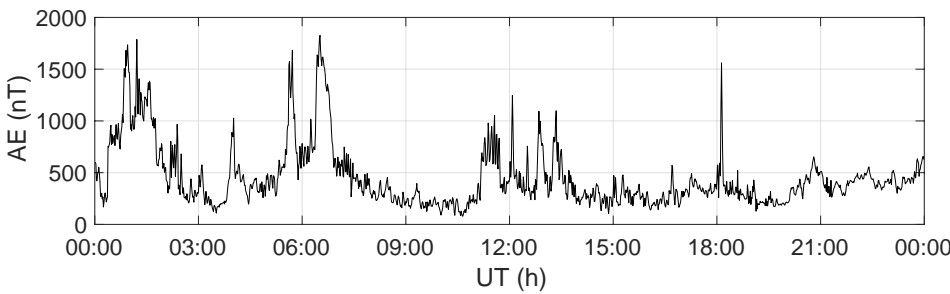

**Figure 1.** The Auroral Electrojet index at 1 min intervals on 31 October 2003.

LSTIDs during the first two days of the October 2003 ionospheric storms have been studied extensively (e.g. Afraimovich et al., 2006; Ding et al., 2007; Perevalova et al., 2008; Valladares et al., 2009; Borries et al., 2009; Horvath and Lovell, 2010).

This study focuses on the less intense third day of the storms, 31 October 2003, and specifically on a high-amplitude TID observed over North America in the local morning hours (16-20 UTC).

Section 2 covers the data instrumentation used for the study of the TID, and shows examples of the GNSS slant TEC (sTEC) observed. In section 3.1, observations from different instruments and techniques - GPS tomography, an ionosonde and a space-borne Planar Langmuir Probe (PLP) are compared. To investigate the effects of using a sparse network of GPS receivers, an

additional tomographic inversion using simulated data is performed in section 4. Section 5 contains a short discussion on the results and generation of the TID and final conclusions.

## 2   Data and instrumentation

The primary data used in this study were sTEC measurements derived from phase delay observations by a network of ground-based dual-frequency GPS receivers. In addition to the GPS sTEC used to image the TID, independent ionosonde data and

measurements from the Challenging Minisatellite Payload (CHAMP) PLP were used to confirm the presence of a TID.



## 2.1 GPS TEC

The GPS receiver network is shown in Figure 2 and includes 40 stations in North America (listed in Table A1) which are part of the International GNSS Service (IGS) and UNAVCO networks.

Slant TEC values were calculated using the geometry-free combination. It should be noted that MIDAS (section 2.1.1) uses
time-differenced sTEC measurements, so satellite- and receiver biases which change slowly over time have no effect on the accuracy of the inversion (Mitchell and Spencer, 2003).

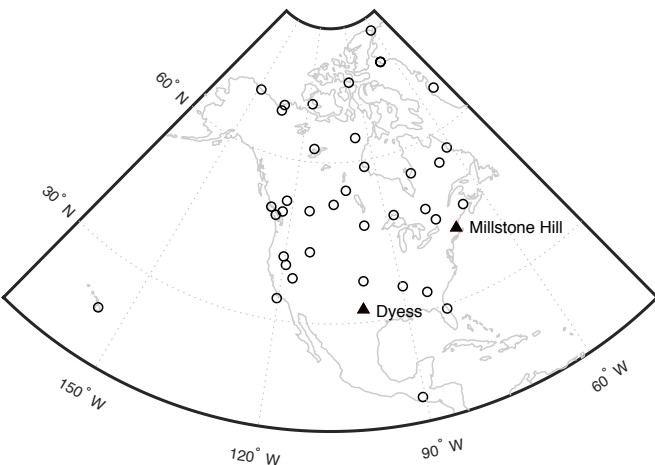

**Figure 2.** Network of GPS receivers used (circles) and the location of the Dyess and Millstone Hill ionosondes (triangles).

Figure 3 shows an example of pseudorange-calibrated sTEC observations from one receiver station, *tono*, where wavelike perturbations can be seen in the sTEC of several satellites. The satellites with the clearest TID signatures, PRNs 3, 28 and, 31, had Ionospheric Pierce Points (IPPs) moving northeast. It should be noted that the movement of the satellites relative to
a TID may result in distortions to the apparent TID, as it introduces a Doppler-like shift in the apparent period of the TID perturbations (e.g. Wan et al., 1997; Hernández-Pajares et al., 2006; Penney and Jackson-Booth, 2015). Bolmgren et al. (2020) showed, using simulations, that MIDAS has the capacity to correctly image LSTIDs without explicitly taking this effect into account.



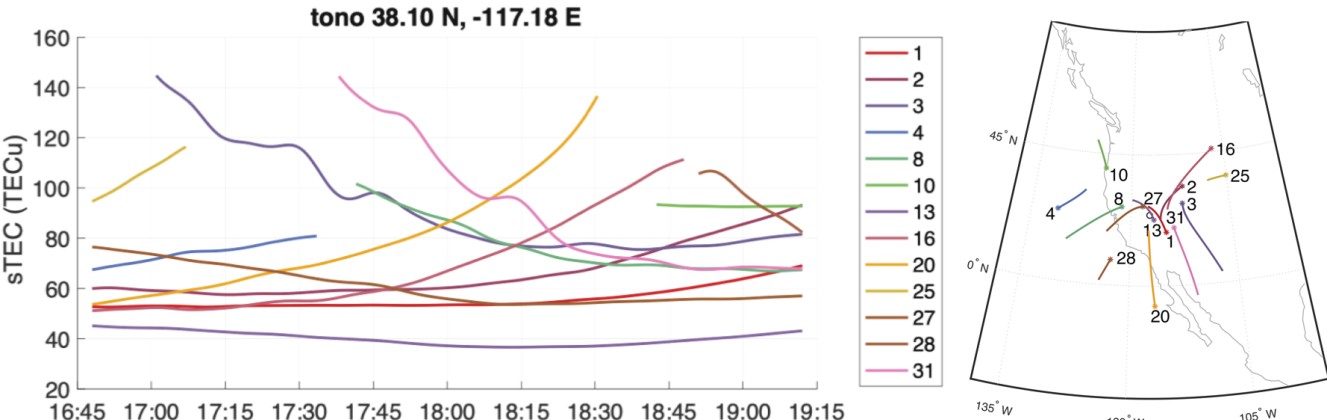

**Figure 3.** Biased sTEC from GPS receiver station tono on 31 October 2003.

### 2.1.1 MIDAS

Computerised ionospheric tomography is a method that can estimate the 2D or 3D ionospheric electron density over an area using integrated electron density measurements, such as TEC. In general, ionospheric tomography can be described as solving an inverse problem formulated by the relationship between the geometry, the observations and the discretised electron density distribution. For a historical review of different methods of ionospheric tomography see Bust and Mitchell (2008).

In this study, the electron density was imaged using the Multi-Instrument Data Analysis Software (MIDAS) tomography algorithm (Mitchell and Spencer, 2003). MIDAS uses differential phase observations from a network of ground-based geodetic GNSS receivers and solves for an estimate of the ionospheric electron density. Empirical Orthogonal Functions (EOFs) are used as a change of basis in the height dimension; this constrains the problem by decreasing the degrees of freedom and by providing a basic structure to the variation of electron density with height. MIDAS has previously been tested as a TID imaging algorithm using a simulation approach in Bolmgren et al. (2020), which established that the algorith can successfully reproduce LSTIDs using GNSS data. In this study we will show that this is possible with real data even in relatively challenging conditions.

### 2.2 Ionosondes

The first scientific observations of TIDs were made using ionosondes (Munro, 1948). Ionosondes are ground based radio instruments that characterise the bottomside electron density of the ionosphere. Ionosondes work by generating signal pulses that sweep through a span of frequencies. The pulses reflected back to the Earth from close to the zenith are used to estimate the height distribution of the plasma frequency, which is proportional to the square root of the electron density, directly above the ionosonde. The highest plasma frequency is usually found in the F2 layer, and is denoted foF2. Since electromagnetic waves with frequencies above foF2 pass through the ionosphere, ionosondes provide no information on the electron density above the height of the F2 layer (referred to as hmF2).





Ionosondes at Dyess (32.4°N, 99.8°W) and Millstone Hill (42.6°N, 71.5°W) were both active on 31 October 2003. Figure
2 indicates the locations of these two ionosondes. The Millstone Hill ionosonde is used as a reference when setting up the
MIDAS EOFs, while measurements from the Dyess ionosonde are used in Section 3.2.

### 2.3 CHAMP Planar Langmuir Probe

The CHAMP satellite was active for ten years between 2000 and 2010, and was equipped with atmospheric and ionospheric
observation instruments. CHAMP has a near circular polar orbit and had an altitude around 390 km at the time of the storm,
which usually would be in the topside of the ionospheric F layer. This study makes use of electron density data from the
CHAMP PLP, a planar langmuir probe which was used to measure in-situ electron temperature as well as electron density in
the front of the spacecraft every 15 s. Details on the CHAMP PLP can be found in McNamara et al. (2007).

## 3 Results

Sections 3.1, 3.2 and 3.3 present the results in terms of the tomographic GPS inversion, foF2 and hmF2 from the Dyess
ionosonde, and CHAMP PLP in-situ electron density respectively.

### 3.1 Tomographic inversion

Differential phase observations **(same)** from the GPS receiver network were used with MIDAS to estimate the ionospheric
electron density distribution on 31 October 2003. The reconstructions in MIDAS used voxels of $2° \times 2° \times 50$ km in latitude,
longitude and height respectively, and time steps of 10 minutes. Two EOFs were generated using a set of Chapman profiles
(Chapman, 1931), adjusted to fit the vertical profiles observed by the Millstone Hill ionosonde.





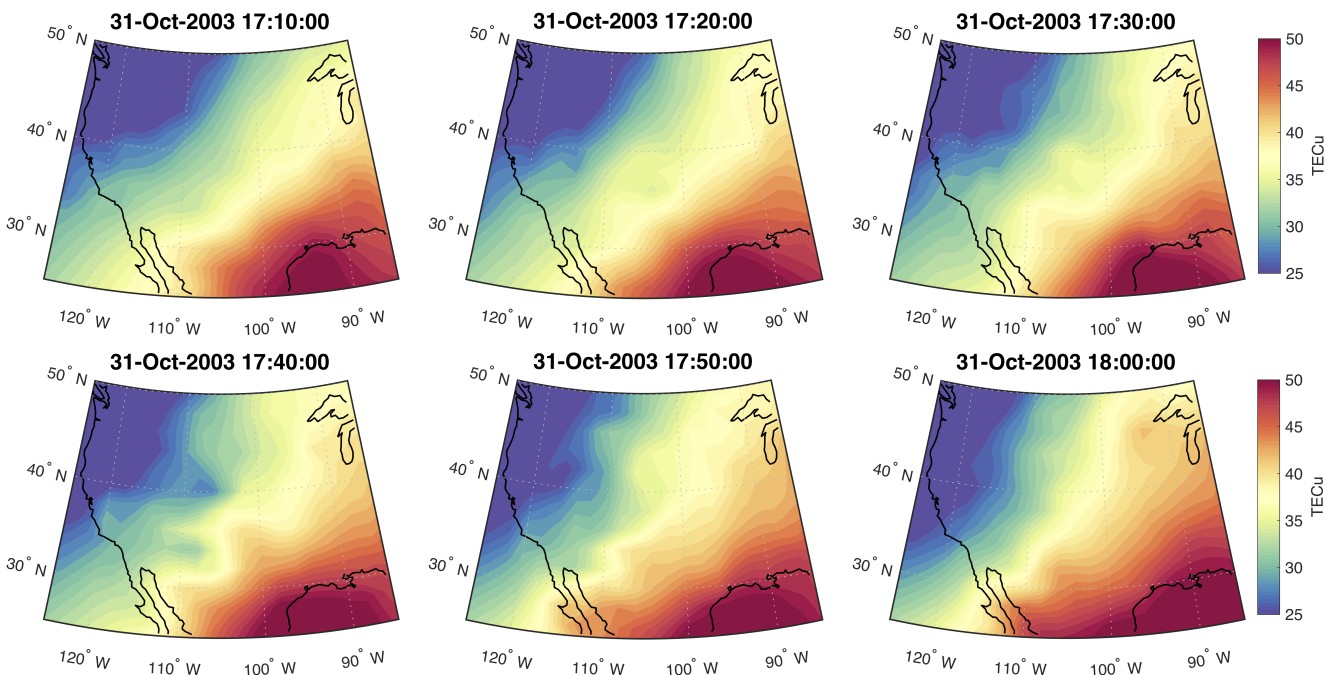

**Figure 4.** Series of vTEC from the MIDAS GPS inversion. Each frame is separated by 10 minutes.

Figure 4 shows six consecutive time frames between 17:10-18:00 of the inversion results, with electron density integrated vertically to give vTEC. Between two and four wave fronts aligned NW-SE can be observed in the figure, spanning latitudes between 45° and 30°. These features are also visible in the electron density viewed as a cross section spanning 100-1200 km in altitude along the direction of travel, shown in Figure 5. The wave-like perturbations are presumed to be the result of a passing TID.





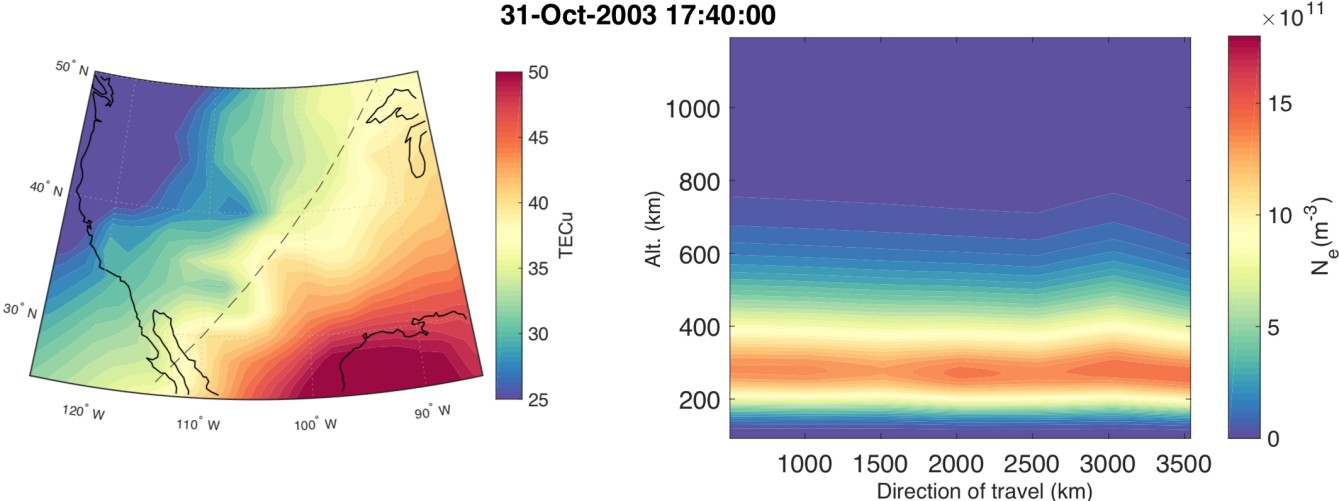

**Figure 5.** (Right) NE-SW cross-section of the full inverted electron density, and (left) the path of the placement of the cross-section in the vTEC map.

Using consecutive tomographic images from MIDAS, the TID parameters were estimated as follows: horizontal wavelength $\lambda_h \approx 700$ km, phase velocity $v_{ph} \approx 390$ m/s, and direction of travel $\approx 195°$ S-W . The period $T$ was estimated as $T = \lambda_h/v_{ph} \approx 30$ min. These parameters would qualify the TID as medium scale, following the definitions in Hunsucker (1982). However, considering the high amplitude, geomagnetic conditions and equatorward direction of travel we will consider it a LSTID.

### 3.2 Ionosonde observations

The Dyess ionosonde is located within the area that was visibly affected by the TID in the MIDAS images. There is an indication of a periodical signature in the F2 layer critical frequency (foF2) with a 30 min period between 18:00 and 19:30 UTC, which may be related to the TID visible in the GPS data. However, the 15 min sampling makes it impossible to detect potential shorter perturbations. In Figure 6, foF2 and hmF2 from the Dyess ionosonde are plotted against the equivalent parameters calculated from the MIDAS result. The other ionosonde with data readily available during this period, Millstone Hill (42.6°N, 71.5°W), does not show a similar indication of TID passage. This is expected, since it is located outside of the area visibly affected by the TID in the tomographic inversions.

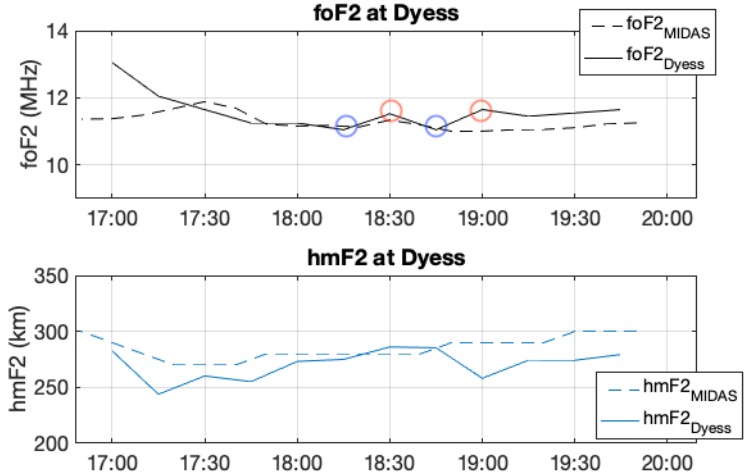

**Figure 6.** foF2 (top) and hmF2 (bottom) observations for the Dyess ionosonde (32.4°N, 99.8°W) and MIDAS equivalent sampled at the same location on 31 October 2003.

### 3.3 CHAMP PLP observations

The CHAMP satellite had one north-to-south pass over North America between 17:00 UTC and 19:00 on 31 October 2003, when the TID was visible in the GPS TEC. The in-situ electron density measured by the PLP at altitudes between 391-395 km for this pass over North America is plotted in Figure 7. At 17:43 UTC, CHAMP passes North America at longitude 76°W, i.e. East of the area where the TID is visible in the tomographic images. Two dips in electron density separated by an apparent latitudinal wavelength of around 700-825 km are visible in Figure 7 between latitudes 15°N and 30°N.

The dotted line in Figure 7 shows the electron density estimated by MIDAS at 17:40 UTC, sampled at the location of CHAMP. Apart from not displaying the same wave perturbations, the electron density at this altitude is overestimated by approximately $3 \times 10^{11}$ electrons per m$^3$. This is the result of a mismatch between the in-situ observation and integrated estimate of the vertical density distribution in this area.

The perturbations in Figure 7 may indeed be caused by the passage of the TID seen further west in the tomographic images, 125 but poor receiver coverage in the region may explain why the wavefronts do not appear to reach 76°W in the tomography result. The effect of possible poor data coverage is further examined by testing the tomography procedure on simulated data in Section 4.

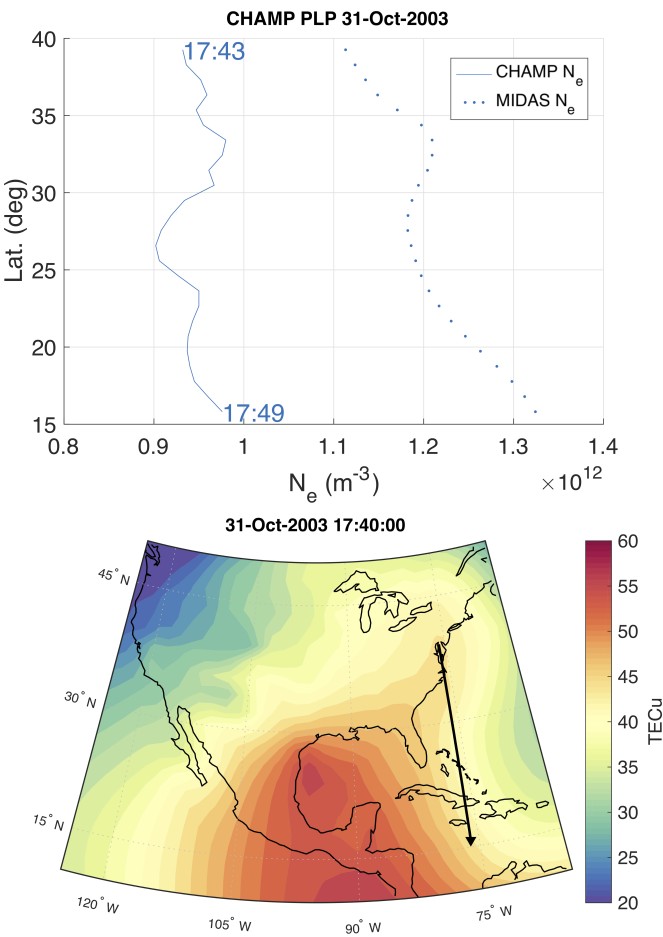

**Figure 7.** in-situ electron density (top) measured by the CHAMP Planar Langmuir Probe (line) and the electron density sampled from the MIDAS inversion (dotted line) for 31 October 2003. Corresponding CHAMP satellite track is plotted on top of the MIDAS vTEC result for 17:40 UTC (bottom).

## 4   Method verification by simulation

The Dyess ionosonde and CHAMP PLP electron density both suggest the presence of a TID, but the wave-like features observed by these instruments are not clearly translated onto the same spatial and temporal coordinates in the MIDAS inversion results. The ionosonde suggests the presence of a TID with a period similar to that in the GPS inversion, but it appears later than it does in the inversion. The CHAMP satellite measurements suggest wave-like perturbations can affect the electron density as high up as 390 km in a region where the wave is not visible in the tomographic inversion. It is possible that these features are not visible in the tomographic images due to poor receiver coverage below 30 °latitude.





To investigate the effect of data-coverage and geometry used for the tomographic inversion, simulated TEC from a model ionosphere was inverted with MIDAS under the same geometric conditions (satellite geometry and receiver coverage) as the original inversion. Any discrepancies between the model and simulated inversion results can be used to identify where there may be issues in the results presented in Section 3.1. A second inversion of the simulated data, using a denser, fictional network is used to identify the effect of receiver geometry.

The TID parameters estimated in section 3.1 were used together with the Hooke (1968) TID model and the International Reference Ionosphere, IRI2016 (Bilitza et al., 2017), to generate a model ionosphere with TID, through which sTEC measurements were integrated (following Bolmgren et al. (2020)). A single frame of the model ionosphere is shown in Figure 8a).

The resulting inversion shows that the while the reconstruction with the regular network (Figure 8b) is able to conserve the
main morphology of the TID, it does not correctly replicate the perturbations of the wave East of 105°W and South of 30°N. In addition, the wavefronts in Figure 8b) appear skewed when compared to the model in Figure 8a). In all panels of Figure 8, a 1 h running mean was subtracted from each voxel post-inversion to minimise the background ionosphere and to better see the TEC perturbations caused by the modelled TID.

The wave is more accurately reproduced if a denser network of GPS receivers than was available in 2003 is used. Figure 8c)
shows the improved simulation result, which uses a larger number of receivers. The simulated receiver network is marked by points in the same sub-figure. This inversion more accurately reproduces the perturbations in Figure 8a), including the direction of the wavefronts.

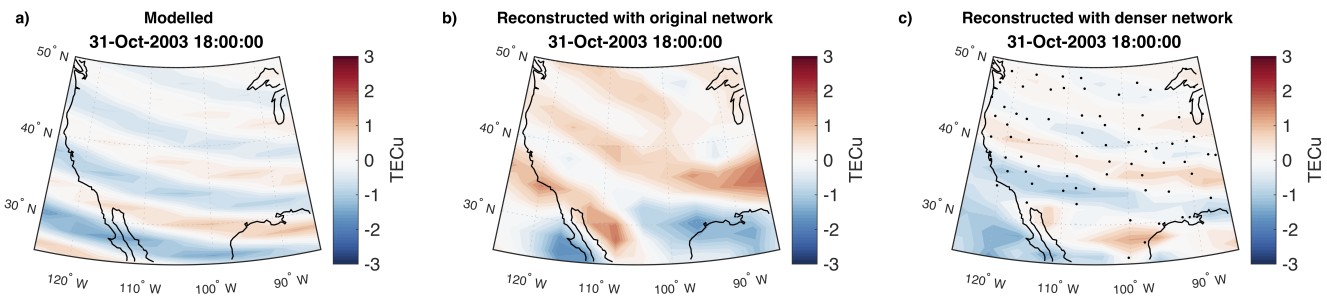

**Figure 8.** a) The modelled LSTID vTEC a and filtered inversions from the simulated data (b) and c). c) uses a denser network than the real inversion, where the receiver locations of the network are indicated with black markers.

## 5   Discussion and Conclusions

In this paper, we have used GPS tomography to reconstruct the ionospheric electron density over North America for 31 October
2003, the third day of the Halloween storm of 2003, and to identify a LSTID. The presence of a large-scale TID was evidenced by other instrumentation. A potential discrepancy in the TID morphology was observed between the measurements of two other instruments and the large-scale MIDAS reconstructions, in particular in that the TID was captured by the Dyess ionosonde and





CHAMP PLP. However, this was identified to be the result of poor receiver coverage available for the MIDAS inversion and was studied through a computer end-to-end simulation, as discussed in Section 5. The receiver network used has an approximate

receiver density of 1 per 10 deg$^2$, compared to approximately 6 per 10 deg$^2$ for the denser synthetic network shown in Figure 8c).

The observed TID had an estimated phase velocity of 390 m/s, an estimated period of 30 min, horizontal wavelength of 700 km and a southwesterly direction, suggesting a source in the auroral region. The high variability in AE occurring between 11:00 UTC and 14:00 UTC (Figure 1) may indicate a possible time of launch of the observed LSTID, if it were launched by

Joule heating resulting from variations in the auroral electrojets. Another possible source mechanism may have been heating by auroral particle precipitation. The auroral oval was centered at latitude 63°N at this time with the region experiencing strong energetic particle precipitation at 14:30 UTC, as estimated by OVATION (Newell et al., 2002) as shown in Figure 9. The highest levels of precipitation around the presumed launch time of the LSTID ocurred between 08:00-10:00 Magnetic Local Time (MLT), which coincides with northern North America at the presumed launch time of the LSTID (11:00-14:00 UTC) and

with the increased levels of AE activity around the same time. However, further analysis of additional datasets would be needed to obtain a detailed understanding of the generation mechanisms responsible for this LSTID. Since TIDs are effectively relative changes in the background electron density, the enhanced storm density likely contributed to the high perturbation amplitudes.

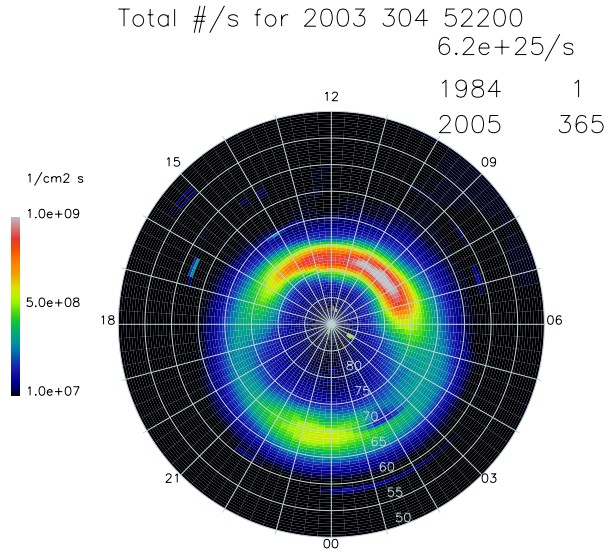

**Figure 9.** OVATION particle flux for 31 October 2003 at 14:30 UTC.

The work discussed in this paper is built on that of Bolmgren et al. (2020), where MIDAS was demonstrated to be capable to image certain TIDs, and has shown that the tomographic algorithm is capable of imaging LSTIDs with relatively small spatial

dimensions, provided that a sufficiently dense ground receiver network is available.





Tomographic maps like the ones produced here could be used in practical navigation systems to provide ionospheric delay corrections for GNSS positioning during LSTID activity, which may otherwise induce unmodelled TEC fluctuations impairing the quality of the solution.

## Appendix A: List of receiver stations

| ID | lat (deg) | lon (deg) | ID | lat (deg) | lon (deg) | ID | lat (deg) | lon (deg) | ID | lat (deg) | lon (deg) |
|---|---|---|---|---|---|---|---|---|---|---|---|
| alrt | 82.4939 | -62.34179 | garl | 40.4165 | -119.3555 | modb | 41.9023 | -120.3028 | sg00 | 47.9218 | -97.08662 |
| bake | 64.3178 | -96.00235 | gtrg | 43.2441 | -113.2412 | nain | 56.537 | -61.68864 | thu2 | 76.537 | -68.82508 |
| bcov | 50.5443 | -126.8426 | guat | 14.5904 | -90.52018 | ormd | 29.2982 | -81.10889 | thu3 | 76.537 | -68.82508 |
| bogt | 4.65553 | -74.10725 | holm | 70.7363 | -117.7613 | prds | 50.8714 | -114.2935 | tono | 38.0972 | -117.184 |
| cags | 45.5851 | -75.80731 | hvlk | 37.6515 | -99.10675 | ptal | 49.2563 | -124.8609 | tukt | 69.4383 | -132.9944 |
| chur | 58.7591 | -94.08873 | invk | 68.3062 | -133.527 | qaq1 | 60.7153 | -46.04779 | vald | 48.097 | -77.56419 |
| cvms | 35.5414 | -89.64351 | kely | 66.9874 | -50.94485 | reso | 74.6911 | -94.8961 | wslr | 50.1265 | -122.9211 |
| dsl1 | 70.3334 | -148.4728 | kuuj | 55.2784 | -77.74545 | ross | 48.8337 | -87.5196 | yell | 62.4809 | -114.4806 |
| flin | 54.7256 | -101.978 | mig1 | 34.0383 | -120.3514 | sask | 52.1963 | -106.3984 | ztl4 | 33.3797 | -84.29673 |
| frdn | 45.9335 | -66.65992 | mkea | 19.8018 | -155.456 | sch2 | 54.8321 | -66.83255 | will | 52.2369 | -122.1679 |

**Table A1.** North American GPS receiver stations used for the tomographic inversion.

**Acknowledgments**

This work was supported by funding from Horizon 2020 Marie Skłodowska-Curie Actions grant agreement No. 722023. Supported by NERC grant number NE/P006450/1. The authors thank the UML DIDBase (http://umlcar.uml.edu/DIDBase) for providing the data from the Dyess and Millstone Hill ionosonde stations. We thank the IGS (http://www.igs.org/) and UNAVCO (https://www.unavco.org/) for providing the The GPS RINEX files, and the WDC for Geomagnetism, Kyoto (http://wdc.kugi.kyoto-185  u.ac.jp/wdc/Sec3.html) for providing the AE index data.





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
