# Peer review of "Tomographic Imaging of a Large Scale TID during the Halloween Storm of 2003"

_Annales Geophysicae, 2020_

## Referee Comment (RC1) · Richard Fallows (Referee) · 15 Jun 2020

The paper summarises the detection of a TID during the "Halloween storms" of October 2003 using a tomographic inversion of received GPS signals to produce 3-D maps of electron density in the ionosphere. These are compared with supporting data from two ionosondes and the CHAMP satellite which was observing at the time. The paper neatly describes the possibilities and limitations of using GPS tomography to detect TIDs and, inadvertently, the limitations of the supporting data available at that time. It is well-written and should be published subject to the, mostly, minor corrections listed below.

Line 23: Remove the phrase "travelling along the interplanetary magnetic field". It is,

[Figure]

at the very least, overly simplistic and a statement that is not needed in this sentence.

Line 69: algorith → algorithm.

Line 92: What is meant to be in the brackets where "same" is written?

Figure 4: The possible TID wavefronts that the authors note are not immediately obvious in this figure without the reader looking very carefully. It would be very helpful to the reader if a 1-hour running mean is subtracted from each voxel to remove the background ionosphere, as performed for the simulations given in Figure 8.

Figure 6: As the authors note in the text, the 15 minute sampling period of the ionosonde data make it impossible to find periodicities of less than 30 minutes, and 30 minutes itself is right on the limit for Nyquist sampling of a wave with this period. The figure itself does not show very clearly even a 30-minute periodicity. Perhaps it might be seen more clearly if the y-axis limits are reduced (e.g., to 10-13MHz or even 10.5-12.5MHz) to better show this. I guess the red and blue circles denote peaks and troughs respectively, but this is not described in the figure caption.

Figure 8: The caption should also note that a background subtraction was performed, otherwise use of the vTEC term is not strictly accurate as it implies absolute values.

Lines 156-158: This sentence is not clear: It states that a discrepancy is observed, but not the nature of that discrepancy.

Lines 158-159: I suggest re-arranging this sentence to state, "However, this was identified from computer end-to-end simulation to be the result of poor receiver coverage available for the MIDAS inversion, as discussed in Section 5."

Line 160: It might be useful to indicate at this point how the coverage of GPS receivers has improved since 2003, and whether it is now dense enough to be able to replicate the presence of a TID more accurately.

---

## Referee Comment (RC2) · Anonymous Referee #2 · 17 Jul 2020

The submitted article is dedicated to the tomographic imaging of a particular large-scale TID occurred in the last day of the Halloween Storm of 2003, namely on October 31st. The authors studied the TID by means of the MIDAS tomography algorithm based on North American GPS receiver network data together with observational data from two ionosondes (Millstone Hill and Dyess) and the CHAMP satellite. I support an approach for fully or partly observational works to investigate events by fusing multi-measurement data. Therefore, I encourage the work done by the authors.

The authors note that the present paper is built on the publication of Bolmgren et al. (2020), where the MIDAS was used to imagine TIDs. Thus, the present work can be considered as another test for the MIDAS. It was demonstrated explicitly a validity of the MIDAS as a technique for searching and visualization of large-scale TIDs, that

works effectively for sparse GPS receiver network. Moreover, it has been shown by simulation that the accuracy of the MIDAS raises noticeably with increasing number of receivers.

The reviewed article brings confirmation of developed earlier technique for imaging of TIDs. In addition, the work shows its novelty and actuality in the view of present-day attention to the deployment of dual-frequency GPS receivers for studies of processes in the Earth's ionosphere. I recommend the manuscript for publication in the Annales Geophysicae with minor revision.

Minor remarks:

Line 22: "...a series of large Coronal Mass Ejections...". The word "large" should be removed, since it brings a comparative feature, that is not needed in the context.

Line 92: The word in bold "...(same)..." should be explained, since it is not clear what is meant.

Figure 3: The sTEC curves corresponding to PRNs 27 and 28 have the same brown colour that makes it difficult for a reader to distinguish them. The authors mentioned about the TIDs signatures in PRN 28 (Line 53). So, it is better to change the colour.

Figure 6: There are two plots showing foF2 and hmF2. The data were taken from the Dyess ionosonde observations and obtained with the MIDAS algorithm. Although, the MIDAS' tracks of foF2 and hmF2 demonstrate similar behavior to those Dyess' corresponding tracks, there are noticeable deviations. In this connection, what is the accuracy of determining foF2, hmF2 and Ne in the MIDAS algorithm with that set of GPS receivers? If the accuracy as a quantitative value can be obtained from the modeling presented in Section 4? The authors may consider mentioning about the algorithm's accuracy in the text.

---

## Author Comment (AC1) · 10 Aug 2020

We would like to thank you for your careful reading of the manuscript and valuable comments and suggestions. Please find our replies to individual comments below.

**Line 23: Remove the phrase "travelling along the interplanetary magnetic field". It is, at the very least, overly simplistic and a statement that is not needed in this sentence.**

We agree that the sentence is not needed and it has been removed.

**Line 69: algorith - algorithm.**

Corrected.

[Figure]

**Line 92: What is meant to be in the brackets where "same" is written?**

This was left in the text by oversight, and has been removed.

**Figure 4: The possible TID wavefronts that the authors note are not immediately obvious in this figure without the reader looking very carefully. It would be very helpful to the reader if a 1-hour running mean is subtracted from each voxel to remove the background ionosphere, as performed for the simulations given in Figure 8.**

A clearer indication of the wavefronts would indeed be useful. We find that the full ionospheric TEC, especially in the areas sampled by the CHAMP PLP and the Dyess ionosonde, are important to keep visible. Therefore we have added arrows on top of the TEC maps in Fig. 8 to indicate the main wave features.

**Figure 6: As the authors note in the text, the 15 minute sampling period of the ionosonde data make it impossible to find periodicities of less than 30 minutes, and 30 minutes itself is right on the limit for Nyquist sampling of a wave with this period. The figure itself does not show very clearly even a 30-minute periodicity. Perhaps it might be seen more clearly if the y-axis limits are reduced (e.g., to 10-13MHz or even 10.5-12.5MHz) to better show this. I guess the red and blue circles denote peaks and troughs respectively, but this is not described in the figure caption.**

The y-axis has been reduced, and this reduced the need for red and blue circles that have been removed.

**Figure 8: The caption should also note that a background subtraction was performed, otherwise use of the vTEC term is not strictly accurate as it implies absolute values.**

Corrected.

**Lines 156-158: This sentence is not clear: It states that a discrepancy is ob-**

**served, but not the nature of that discrepancy.**

This is indeed not clear. The following sentence has been added: "While indications of the TID was captured by the Dyess ionosonde and CHAMP PLP, this was in areas where the MIDAS reconstruction showed no clear wave pattern"

**Lines 158-159: I suggest re-arranging this sentence to state, "However, this was identified from computer end-to-end simulation to be the result of poor receiver coverage available for the MIDAS inversion, as discussed in Section 5."**

The sentence has been re-arranged as suggested.

**Line 160: It might be useful to indicate at this point how the coverage of GPS receivers has improved since 2003, and whether it is now dense enough to be able to replicate the presence of a TID more accurately.**

This is a very relevant point. The following sentence has been added: "For comparison, the modern North American network used by Bruno et al. (2020) has an average receiver density close to 15 per 10x10 deg."
* * *
**foF2 at Dyess**

**hmF2 at Dyess**

**Fig. 1.**

[Figure]

**Fig. 2.**

---

## Author Comment (AC2) · 10 Aug 2020

We would like to thank you for your careful reading of the manuscript and valuable comments and suggestions. Please find our replies to individual comments below.

**Line 22: ". . .a series of large Coronal Mass Ejections. . .". The word "large" should be removed, since it brings a comparative feature, that is not needed in the context.**

The word has been removed.

**Line 92: The word in bold "...(same)..." should be explained, since it is not clear what is meant.**

[Figure]

This was left in the text by oversight, and has been removed.

**Figure 3: The sTEC curves corresponding to PRNs 27 and 28 have the same brown colour that makes it difficult for a reader to distinguish them. The authors mentioned about the TIDs signatures in PRN 28 (Line 53). So, it is better to change the colour.**

This is an important observation. The colours have been changed.

**Figure 6: There are two plots showing foF2 and hmF2. The data were taken from the Dyess ionosonde observations and obtained with the MIDAS algorithm. Although, the MIDAS' tracks of foF2 and hmF2 demonstrate similar behavior to those Dyess' corresponding tracks, there are noticeable deviations. In this connection, what is the accuracy of determining foF2, hmF2 and Ne in the MIDAS algorithm with that set of GPS receivers? If the accuracy as a quantitative value can be obtained from the modeling presented in Section 4? The authors may consider mentioning about the algorithm's accuracy in the text.**

Some text and a reference regarding this accuracy has been added: "In Table 4 of Bruno et al. (2020), MIDAS results were compared against ionosonde data, and for a setup close to what is used here Bruno et al. (2020) found errors of 0.55 MHz in foF2 and 40 km in hmF2. The discrepancies in Figure 6 are on the same order."
* * *
[Figure]

**Fig. 1.**